# In-situ physical adjoint computing in multiple-scattering electromagnetic environments for wave control

John Guillamon[1,3], Cheng-Zhen Wang [1,3], Zin Lin[2] & Tsampikos Kottos [1] ✉

Controlling electromagnetic wave propagation in multiple scattering systems is a challenging endeavor due to the extraordinary sensitivity generated by strong multi-path contributions at any given location. Overcoming such complexity has emerged as a central research theme in recent years, motivated both by a wide range of applications – from wireless communications and imaging to optical micromanipulations – and by the fundamental principles underlying these efforts. Here, we show that an in-situ manipulation of the myriad scattering events, achieved through time- and energy-efficient adjoint optimization (AO) methodologies, enables real time wave-driven functionalities such as targeted channel emission, coherent perfect absorption, and camouflage. Our paradigm shift exploits the highly multi-path nature of these complex environments, where repeated wave-scattering dramatically amplifies small local AO-informed system variations. Our approach can be immediately applied to in-door wireless technologies and incorporated into diverse wave-based frameworks including imaging, power electronic and optical neural networks.

Controlling electromagnetic wave propagation in naturally occurring or engineered multi-mode complex media is a core challenge for RF/microwave, modern optical, and photonic systems[1–17]. The origin of this difficulty lies in multiple scattering and the consequent interference of many photon paths, leading to extraordinary complexity and sensitivity in these media. Yet, controlling these wave-scattering events and their associated interference phenomena is essential for a wide range of applications, including satellite and in-door wireless communications, fiber-based communications and endoscopy, deep-tissue imaging, and optogenetic control of neurons. At first glance, the complete scrambling of a wavefront as it propagates through a complex medium appears to conflict with the objective of precision wave-control—such as focusing electromagnetic radiation on a diffraction-limited spot inside or through a multi-scattering/opaque medium. Indeed, for many years, the presence of random secondary sources (scatterers or reflectors) was considered detrimental. However, novel

techniques such as time-reversal[18,19] and wavefront shaping (WS)[1,2,19] disrupted this paradigm by recognizing that these secondary sources offer additional degrees of freedom. Wavefront shaping protocols have relied on recent technological developments with spatial light/microwave modulators[19–22]; these allow phase and/or amplitude modulation to each segment of an incident monochromatic wavefront in order to achieve desired functionalities after propagation through the complex medium. On the other hand, time-reversal provides a broadband approach that yields spatiotemporal focusing of waves.

Although both of these methodologies guarantee optimal efficiency, they require a complete knowledge of the scattering domain, limiting their practicality for a variety of applications. A pivotal example is indoor wireless communications[4,23,24], where small temporal variations in the enclosure can drastically alter the scattering process. An entirely different approach relying on smart electromagnetic environments has emerged with the advent of

[1]Wave Transport in Complex Systems Lab, Department of Physics, Wesleyan University, Middletown, CT, USA. [2]Bradley Department of Electrical and Computer Engineering, Virginia Tech, Blacksburg, VA, USA. [3]These authors contributed equally: John Guillamon, Cheng-Zhen Wang.
✉e-mail: tkottos@wesleyan.edu

reconfigurable intelligent metasurfaces (RISs)[23–29]. This approach foresees a fully programmable wave propagation to harness the wave-scattering complexity and achieve optimized transmission of both information and power. A bottleneck for the practical implementation of this proposal is the development of smart, low-cost/high-efficiency, optimization schemes that will be able to identify in real-time, with low latency, optimal RIS configurations for achieving specific modalities.

Meanwhile, there has been widespread attention towards physical (optical) analog computing for low-latency deep learning applications[30–38]. Pioneering works such as[36,37] showed that it is possible to perform in-situ backpropagation through a photonic implementation of an artificial neural network. However, these platforms typically consist of feed-forward waveguides, couplers and interferometers, in contrast to complex multi-scattering multi-resonant electromagnetic environments, in which real-time optimizations of on-demand wave-control functionalities are needed. Obviously, the development of such real-time, on-hardware optimization schemes will benefit not only indoor wireless communications (see Fig. 1a) but also a broad class of physical systems. Representative wave settings include control of closed-loop tabletop experiments, seismic wave control, as well as applications in adaptive optics (atmospheric imaging, endoscopy, free-space optical links) and related remote-sensing tasks – including geo-satellite imaging and adaptive ranging (optical or RF), see Fig. 1b,c.

Here, we experimentally demonstrate an in-situ Physical Adjoint Computing (iPAC) optimization protocol that leverages adjoint sensitivity analysis[39–42] to control and harness complex wave dynamics. The protocol is built around three components: in-situ measurements, targeted perturbations, and an external control mechanism, which collectively enables the real-time optimization of wave systems through two sequential field propagations – forward and adjoint. First, local probes are employed to measure both the forward and adjoint wave fields at specific elements within the system. Forward propagation is utilized to compute a desired merit (or cost) function and to determine the excitation profile required for the subsequent adjoint propagation. The adjoint field, in turn, provides a comprehensive and simultaneous measurement of all targeted sensitivities. An external control mechanism evaluates these sensitivities to identify the potential perturbations that could enhance (or diminish) the merit (cost) function. These adjustments are then delivered by local actuators to the targeted components (i.e., the tunable degrees). The cycle is repeated as many times as necessary to maximize (minimize) the merit (cost) function. The proposed methodology typically achieves the fastest local convergence among competing strategies, which constitutes it attractive for a variety of wave-control applications, see Fig. 1a–c. While the ultimate gain is context-dependent, iPAC is most effective in controlled, quasi-static or slowly varying, multiply scattering environments where forward and adjoint excitations can be launched. Provided adequate sensing-actuation bandwidth, signal-to-noise ratio, and tolerable local model mismatch, iPAC provides a scalable path to on-platform optimization. To demonstrate the versatility of our protocol we showcase three different modalities, namely, targeted channel emission, coherent perfect absorption, and camouflaging, using a microwave experimental platform, see Fig. 1d. The latter consists of a network of coupled coaxial microwave cables whose wave transport demonstrates features characterizing wave chaotic

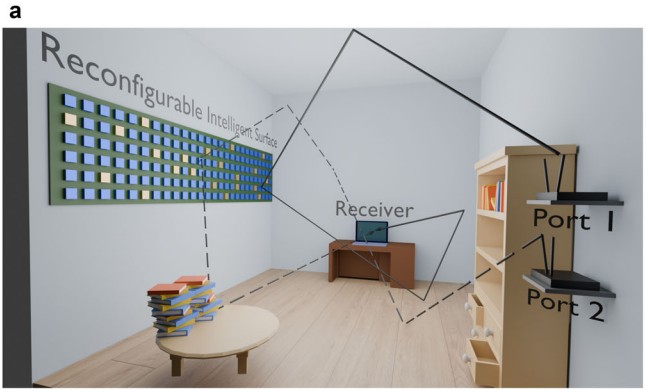

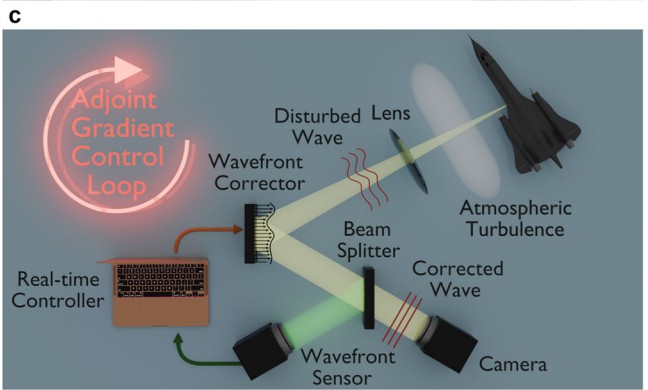

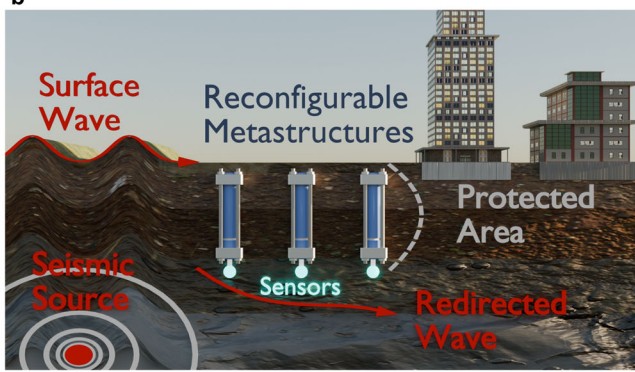

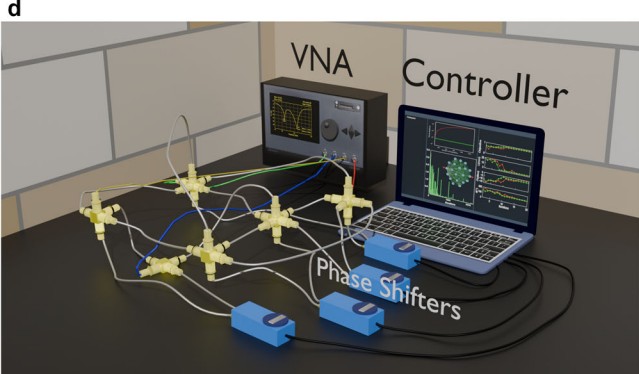

**Fig. 1 | Physical frameworks and experimental platform used to demonstrate iPAC.** Examples of physical wave frameworks where the iPAC protocol can be sucessfully implemented: **a** A smart electromagnetic environment utilizing RIS for a variety of modalities. **b** Seismic wave management. **c** Adoptive optics for atmospheric imaging. **d** A multi-resonant, multi-scattering complex network of coaxial cables has been used as a platform to demonstrate the viability of the iPAC protocol. The protocol used as control parameters, the relative amplitudes and phases of injected waves (which were controlled by the VNA), and a targeted set of coaxial cables of the network whose electrical lengths were digitally controlled using phase-shifters. The iPAC scheme has been demonstrated for a variety of modalities: (i) Targeted Mode Transfer that aims to receive the injected electromagnetic signal at a specific receiving channel; (ii) Coherent Perfect Absorption that aims to absorb the injected electromagnetic signal completely; (iii) Invisibility (cavity camouflaging) that results in an outgoing signal being the same (phase and amplitude) as the injected one.

systems[43–45]. These networks are frequently used as models for mesoscopic quantum transport, sound propagation, and electromagnetic wave behavior in complex interconnected structures such as buildings, ships, and aircrafts[46–50] and therefore constitute a versatile platform for experimentally implementing our in-situ optimization protocol.

## Results

### Principles of in-situ adjoint optimization

Formally, the steady-state propagation of a time-harmonic wave field $\Phi$ is governed by a linear system: $\mathbf{M}(\mathbf{p})\Phi = \mathbf{b}(\mathbf{p})$. Here, $\mathbf{M}$ is the system matrix, $\mathbf{b}$ is the driving source, and $\mathbf{p}$ is a vector of $N$-controllable optimization parameters. An optimization objective $g$ is typically expressed as an explicit function of $\Phi$, $\Phi^*$, and $\mathbf{p}$, i.e., $g = g(\Phi, \Phi^*, \mathbf{p})$. Using the chain rule, the gradient sensitivities of $g$ with respect to $\mathbf{p}$ are given by:

$$\frac{dg}{d\mathbf{p}} = \frac{\partial g}{\partial \mathbf{p}} + 2\mathcal{R}\left(\frac{\partial g}{\partial \Phi}^T \mathbf{M}^{-1}\left(\frac{\partial \mathbf{b}}{\partial \mathbf{p}} - \frac{\partial \mathbf{M}}{\partial \mathbf{p}}\Phi\right)\right) \quad (1)$$

Here, $\frac{\partial \mathbf{b}}{\partial \mathbf{p}} - \frac{\partial \mathbf{M}}{\partial \mathbf{p}}\Phi$ physically represents a collection of induced excitations resulting from perturbing the system via one parameter $p_i$ at a time for each $i = 1, 2, 3, \cdots, N$. Consequently, $\mathbf{U} = \mathbf{M}^{-1}\left(\frac{\partial \mathbf{b}}{\partial \mathbf{p}} - \frac{\partial \mathbf{M}}{\partial \mathbf{p}}\Phi\right)$ denotes a collection of several wave fields in response to each and every one of these perturbations (the $i$th column, $U_i$, corresponds to the wave field generated by perturbing the single $p_i$). However, finding the entire $\mathbf{U}$ becomes excessive especially when the number of controllable parameters, $N$, is large.

The adjoint method addresses this challenge by reformulating the problem as:

$$\left(\frac{\partial g}{\partial \Phi}\right)^T \mathbf{M}^{-1} = \Psi^T \Rightarrow \mathbf{M}^T\Psi = \frac{\partial g}{\partial \Phi} \quad (2)$$

Here, $\Psi$ is the adjoint field generated in response to the source $\frac{\partial g}{\partial \Phi}$. For reciprocal wave media, where $\mathbf{M}^T = \mathbf{M}$, the adjoint field can be found by propagating through the same system. The gradient sensitivities are now given by:

$$\frac{dg}{d\mathbf{p}} = \frac{\partial g}{\partial \mathbf{p}} + 2\mathcal{R}\left(\Psi^T\left(\frac{\partial \mathbf{b}}{\partial \mathbf{p}} - \frac{\partial \mathbf{M}}{\partial \mathbf{p}}\Phi\right)\right) \quad (3)$$

This formulation significantly reduces computational demands, as all sensitivities can be obtained through a single additional field propagation, instead of computing the entire collection $\mathbf{U}$ of $N$ wave fields. Moreover, $\frac{\partial \mathbf{b}}{\partial \mathbf{p}}$ and $\frac{\partial \mathbf{M}}{\partial \mathbf{p}}$ are typically very sparse tensors since the effect of each parameter $p_i$ on $\mathbf{M}$ and $\mathbf{b}$ is localized. Consequently, only the values of $\Phi$ and $\Psi$ corresponding to the non-zero entries of $\frac{\partial \mathbf{b}}{\partial \mathbf{p}}$ and $\frac{\partial \mathbf{M}}{\partial \mathbf{p}}$ are needed.

To implement the adjoint method experimentally, we sequentially excite the wave system with the driving sources $\mathbf{b}$ and $\frac{\partial g}{\partial \Phi}$, measure $\Phi$ and $\Psi$ at the strategic positions designated by $\frac{\partial \mathbf{b}}{\partial \mathbf{p}}$ and $\frac{\partial \mathbf{M}}{\partial \mathbf{p}}$ and then compute $\frac{dg}{d\mathbf{p}}$ digitally using Eq. (3). Importantly, our in-situ protocol bypasses the computationally intensive tasks of solving $\mathbf{M}\Phi = \mathbf{b}$ and $\mathbf{M}\Psi = \frac{\partial g}{\partial \Phi}$. Instead, we directly measure $\Phi$ and $\Psi$, inherently accounting for all the complexities of the system, including hidden losses and detunings, thereby enabling self-calibration. Having found $\frac{dg}{d\mathbf{p}}$, any gradient-guided optimization algorithm can be applied to advance $g$[51]. We set up an external control enclosure to orchestrate the entire process, including the sequential (forward and adjoint) wave-field excitations, in-situ measurements, gradient computations, and optimization updates, ensuring seamless and efficient real-time optimization.

While adjoint analysis shares a conceptual common ground with the celebrated backpropagation algorithm, our goal is not to develop a

physical deep-learning platform[36]. Instead, we aim to optimize a wave system in-situ to achieve specific physical functionalities in real time, such as perfect absorption, signal delivery to targeted channels, or camouflage. Unlike data-driven methods, our protocol does not train any neural network nor rely on extensive datasets. Crucially, our work should be distinguished from physical implementations of feed-forward neural networks[36,37], which often avoid back reflections. In contrast, our in-situ optimization holistically exploits the intricate physics of multiple scattering of waves within an arbitrarily complex network topology, where any wave effect, including back reflections and even resonant phenomena, can be utilized as valuable physical degrees of freedom. Furthermore, our implementation at RF and microwave frequencies allows us to easily access both phase and amplitude information of the fields, which ensures that the intricate wave interactions within the system are accurately accounted for, enabling precise and reliable optimization.

### Physical platform and implementation of the adjoint protocol

The complex microwave network[14,44,52–54] consists of $n = 1, \cdots, V$ vertices, that are connected by one-dimensional coaxial wires (bonds) $B = (n, m)$ of length $L_B$, which are irrationally related to one another. The position $x_B = x$ on bond $B$ is $x = 0 (l_B)$ on vertex $n(m)$. The connectivity of the network is encoded in the $V \times V$ symmetric adjacency matrix $\mathbf{C}$, with elements $C_{nm} = 1$ if vertices $n \neq m$ are connected via a bond $l_B$, and $C_{nm} = 0$ otherwise. The number of bonds that emanate from a vertex $n$ defines its valency $v_n$. The electric potential difference (voltage) between the inner and outer conductor surfaces of the coax cables at position $x$ along each bond satisfy the telegraph equation[44,52–54]

$$\left(\frac{d^2}{dx_B^2} + k^2\right)\psi_B(x_B) = 0; \quad k = \frac{\omega n_r}{c} \quad (4)$$

where $k$ is the wavenumber of the propagating wave with angular frequency $\omega$, $c$ is the speed of light in vacuum and $n_r$ is the complex-valued relative index of refraction of the coaxial cable whose imaginary part describes Ohmic losses in the cables. To emulate realistic conditions, we have considered that all cables suffer Ohmic losses which are modeled by a complex refractive index with imaginary part $\mathcal{I}m(n_r) \simeq 0.0085$. Furthermore, it is convenient to define the vertex field $\Phi = (\phi_1, \phi_2, \ldots, \phi_N)^\top$ where $\psi_B(x_B = 0) = \phi_n$ is the voltage at vertex $n$.

The scattering set-up is completed by connecting $\alpha = 1, \cdots, N \leq V$ of the vertices to transmission lines (TL) that are used to inject and receive monochromatic waves of angular frequency $\omega = 2\pi f$. The coupling to the TLs is described by the $N \times V$ matrix $\mathbf{W}$ with elements 1(0) when a vertex is connected (not connected) to a TL. At each vertex $n$, the continuity of the field and current conservation are satisfied. In the frequency domain, these conditions take the following compact form[44,52]

$$\left(\mathbf{H}(k) + i\mathbf{W}^\top\mathbf{W}\right)\Phi = \mathbf{b}; H_{nm} = \begin{cases} -\sum_{l \neq n} C_{nl}\cot(kL_{nl}), & n = m, \\ C_{nm}\csc(kL_{nm}), & n \neq m. \end{cases} \quad (5)$$

Above, $\mathbf{b} = 2i\mathbf{W}^\top\mathbf{I}$ is the $N$-dimensional vector that describes the driving source, and $I_\alpha = A_\alpha e^{i\theta_\alpha}$ are the components of an $L$-dimensional vector $\mathbf{I}$ that describes the amplitudes $A_\alpha$ and the phases $\theta_\alpha$ of the input fields $I_\alpha$ from the $\alpha$-lead. Consequently, the system matrix for the microwave network is $\mathbf{M} = (\mathbf{H}(k) + i\mathbf{W}^\top\mathbf{W}) = \mathbf{M}^\top$.

The gradient sensitivities are evaluated using Eq. (3). The implementation of this equation requires the knowledge of the adjoint field $\Psi$ which is the solution of the adjoint Eq. (2). In our case, it takes the same form as the equations that dictate the forward field with the only difference being the driving source vector, i.e., $\mathbf{M}\Psi = 2i\mathbf{W}^\top\left(\frac{\partial g}{\partial \Phi}\right)^\top$. The

latter is determined from the specific form of the optimization objective function $g(\mathbf{\Phi}, \mathbf{\Phi}^*, \mathbf{p})$.

The other elements required for the evaluation of Eq. (3) are the gradients $\frac{\partial \mathbf{M}}{\partial \mathbf{p}}$ and $\frac{\partial \mathbf{b}}{\partial \mathbf{p}}$. The optimization parameter vector $\mathbf{p}$ is partitioned into two parts: the first one involves cavity-shaping optimization parameters (e.g. selected set of bond lengths $\{L_{nm}^{opt}\}$ in the network), which are encoded in the system matrix $\mathbf{M}$. Its gradient $\frac{\partial \mathbf{M}}{\partial L_{nm}^{opt}}$ is a sparse $V \times V$ operator with non-zero elements only at entries that incorporate the selected bonds $\{L_{nm}^{opt}\}$. We also consider additional optimization parameters, i.e., the amplitudes $A_\alpha$ and phases $\theta_\alpha$ of the incident waves injected into the system from the $\alpha$-th TL. These wavefront shaping parameters are encoded in $\mathbf{b}$; resulting in $\frac{\partial b_n}{\partial A_\alpha} = 2i e^{i\theta_\alpha} W_{n,\alpha}$, and $\frac{\partial b_n}{\partial \theta_\alpha} = -2A_\alpha e^{i\theta_\alpha} W_{n,\alpha}$.

Eventually, the objective function gradient becomes:

$$\frac{dg}{d\mathbf{p}} \equiv \left[ \frac{dg}{dL_{nm}^{opt}}, \frac{dg}{dA_\alpha}, \frac{dg}{d\theta_\alpha} \right] = \begin{bmatrix} -2\mathcal{R}\left\{ \mathbf{\Psi}^T \frac{\partial \mathbf{M}}{\partial L_{nm}^{opt}} \mathbf{\Phi} \right\} \\ \frac{\partial g}{\partial A_\alpha} + 2\mathcal{R}\left\{ \mathbf{\Psi}^T \frac{\partial b}{\partial A_\alpha} \right\} \\ \frac{\partial g}{\partial \theta_\alpha} + 2\mathcal{R}\left\{ \mathbf{\Psi}^T \frac{\partial b}{\partial \theta_\alpha} \right\} \end{bmatrix}^\top . \quad (6)$$

It is important to emphasize that Eq. (6) does not require measuring the entire $\mathbf{\Phi}$ or $\mathbf{\Psi}$ but only those voltages that correspond to the non-zero entries of $\frac{\partial \mathbf{M}}{\partial L_{nm}^{opt}}$ and $\frac{\partial b}{\partial A_\alpha}, \frac{\partial b}{\partial \theta_\alpha}$ associated with the controllable parameters. In other words, the sparsity of $\partial \mathbf{M}/\partial \mathbf{p}$ and $\partial \mathbf{b}/\partial \mathbf{p}$ further reduces measurement complexities.

## Examples of modalities and optimization objective functions
Below we provide some examples of optimization objective functions associated with various modalities.

Targeted Mode Transfer—In many practical scenarios, particularly in in-door wireless communications, one requires energy/information transfer from specific input channels to designated output channels—distinct from the injected ones. Such targeted mode transfer (TMT) can be achieved by an appropriate cavity-shaping and/or wavefront-shaping process whose success is quantified by the objective function

$$g_{TMT} = \frac{\sum_{\{T_\alpha\}} |\phi_\alpha|^2}{\sum_{\{I_\beta\}} |A_\beta|^2}, \quad (7)$$

where $\{T_\alpha\}(\neq \{I_\beta\})$ denotes the set of targeted (injected) channels. In case of lossless structures $g_{TMT} = 1(0)$ indicates perfect (poor) TMT performance.

Coherent Perfect Absorption—Coherent perfect absorption (CPA)[14,55,56] requires that the incident radiation has a particular frequency and spatial field distribution (coherent illumination) such that the (weakly) absorbing cavity acts as a perfect constructive interference trap that eventually absorbs completely the incident radiation. The adjoint optimization methodology can be utilized for the management of the multi-path constructive interference via cavity shaping and/or wavefront shaping. In this case, the optimization objective function is

$$g_{CPA} = 1 - \frac{\sum_{\{I_\alpha\}} |\phi_\alpha - A_\alpha e^{i\theta_\alpha}|^2}{\sum_{\{I_\alpha\}} |A_\alpha|^2} - \frac{\sum_{\{T_\beta\}} |\phi_\beta|^2}{\sum_{\{I_\alpha\}} |A_\alpha|^2} \quad (8)$$

where the second term describes the reflected waves from the injected channels $\{I_\alpha\}$ and the third term describes the transmitted waves from the remaining $\{T_\alpha\} \neq \{I_\beta\}$ channels. Perfect absorption corresponds to $g_{CPA} = 1$.

Invisibility – Evading the detectability of a scattering object requires the elimination of any imprints in the phase and amplitude of the scattered interrogating waves due to their interaction with a target. This is achieved by appropriate manipulation of the many-path interference phenomena occurring inside the scattering domain via cavity

shaping and/or tailoring control signals that counter phase and amplitude scattering imprints (including absorption) caused by interactions with the target. The objective function that ensures such optimal cancellations take the form

$$g_{invis} = \frac{|\phi_{\alpha_0} - A_{\beta_0} e^{i\theta_{\beta_0}}|^2}{A_{\beta_0}^2} + \frac{|\phi_{\beta_0} - A_{\beta_0} e^{i\theta_{\beta_0}}|^2}{\sum_{\{I_\beta\}} |A_\beta|^2} + \frac{\sum_{\{T_{\alpha \neq \alpha_0, \alpha_c}\}} |\phi_\alpha|^2}{\sum_{\{I_\beta\}} |A_\beta|^2}, \quad (9)$$

where $g_{invis} = 0$ indicates optimal invisibility/camouflage performance. Above, the first term on the right-hand-side compares the scattered signal (phase and amplitude) from a probed $\alpha_0$-channel to an interrogating signal injected into the system from a $\beta_0$-channel; the second term measures the reflectance from the $\beta_0$-channel. Finally, the last term evaluates the transmittance to all channels that are different from the probe channel $\alpha_0$ and the control channel $\alpha_c$. The objective function Eq. (9) does not enforce any constraints to the reflected wave from the control channel $\alpha_c$.

## In-situ Implementation of iPAC
We proceed with the in-situ implementation of our optimization scheme for the three modalities discussed above. The schematics of the microwave networks for each of the three cases are shown in the upper row of Fig. 2. The TLs (black wiggling lines) are attached to vertices that are indicated with red-filled circles. The amplitude and phase of the injected signals from a two-source VNA, have been used as optimization parameters (wavefront-shaping). In all cases, the signal from the $\alpha = 1$ TL serves as a reference for the amplitude $A$ and the phase $\theta$ of the signal injected from the second TL. Finally, the bond $L_{12}^{opt}$ incorporates a phase-shifter which was digitally controlled for "cavity-shaping" purposes. The selected bond was chosen solely for experimental convenience; it is not special to any of the three modalities (TMT, CPA, invisibility). The only practical requirement is that the bond accommodate a phase shifter and that the fields at its end vertices be accessible for probing. Any bond meeting these conditions would serve equally well. In our prototypes, tuning a single cable length was sufficient to realize the targeted functionalities described in the previous subsection.

The in-situ optimization protocol proceeds as follows: *(a)* Forward Measurement: First, we inject signals from TLs attached to two of the vertices ($n = 1, 2$ for the CPA and $n = 1, 3$ for the TMT and invisibility modalities, see upper row of Fig. 3) into the network. Forward voltages $\mathbf{\Phi}$ are measured at the vertices $n = 1, 2$ that are associated with the length optimization parameter $L_{12}^{opt}$. *(b)* Adjoint Measurement: The source for the adjoint measurement is then constructed from the previously measured forward voltages and according to the specific objective function. For the TMT case, the adjoint input was delivered from the $\alpha = 4$ TL that was targeted for maximizing the outgoing signal. For the CPA protocol, the adjoint input was delivered to the two $\alpha = 1, 2$ TLs. Finally, for the invisibility protocol, the adjoint signal was delivered to $\alpha = 1, 2$ TLs where the optimization constraints have been imposed. In this case, a control field was also injected from the remaining $\alpha = 3$ TL. The adjoint voltages $\mathbf{\Psi}$ that were needed to measure for the evaluation of the gradient were associated with the vertices connected to the TLs that have been used to inject the input signal, i.e. $\alpha = 1, 2$ for the CPA, $\alpha = 1, 3$ for the TMT and invisibility protocols; *(c)* Gradient Calculation: With the local forward $\mathbf{\Phi}$ and adjoint $\mathbf{\Psi}$ measurements obtained, we calculate the gradient of the objective function with respect to the controllable parameters (see Eq. (6)) in real-time. We re-emphasize that the choice of controllable parameters dictates the positions/vertices $n$ where the required forward $\phi_n$ and adjoint measurements $\psi_n$ of the voltages are performed for the gradient calculation $\frac{dg}{d\mathbf{p}}$; *(d)* Parameter Update: Once the gradient was computed, we employed a gradient descent algorithm to update the optimization parameters. Specifically, we used the package

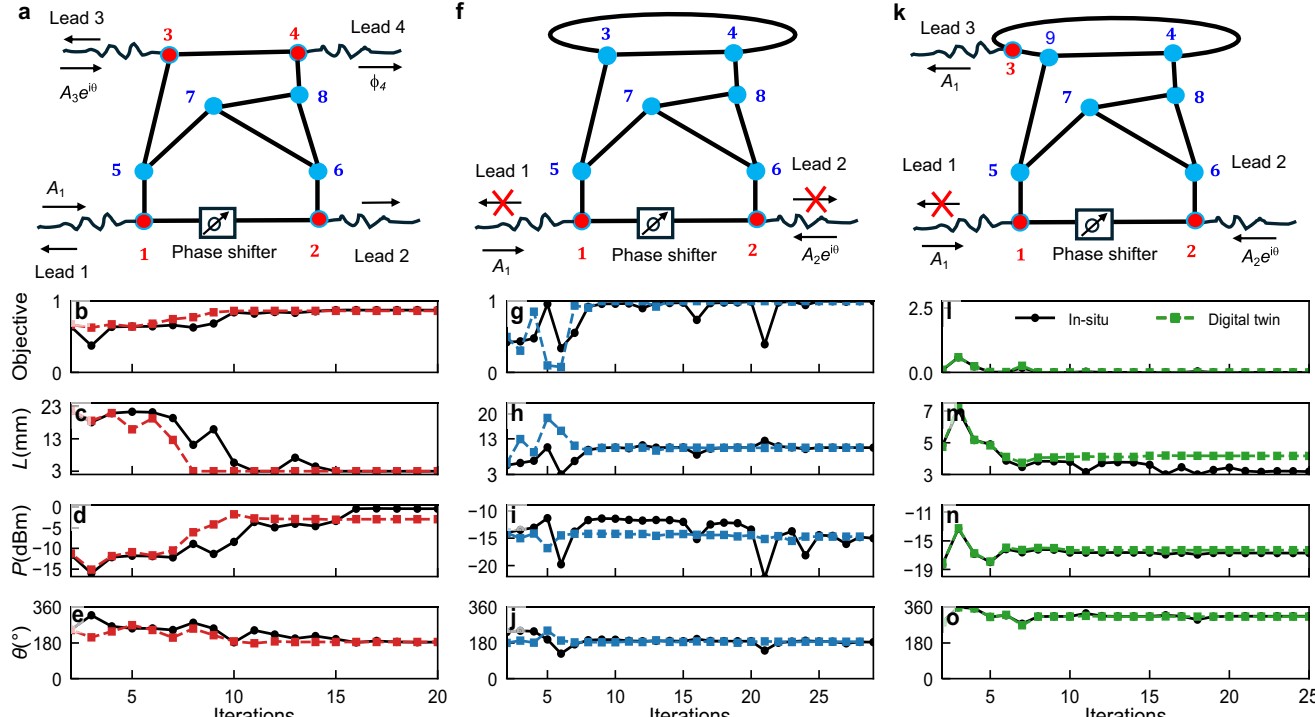

**Fig. 2 | Experimental demonstration of iPAC optimizer.** In-situ demonstration of the iPAC optimizer using a complex network of coaxial cables (red vertices indicate the positions where TLs are attached). The control parameters are the (electrical) length of the cable $L_{12}^{opt}$ (using a phase shifter) and the relative phase and amplitude of the injected signals. **a** Network schematics for Targeted Mode Transmission (TMT) for a wave with frequency $f = 1.86$ GHz that is injected from TLs $\alpha = 1, 3$ and is delivered at targeted TL $\alpha = 4$. **b** Convergence of the TMT objective function $g_{TMT}$ ($g_{TMT} \approx 87\%$) vs. iteration number. Evolution of: **c** The electrical length $L_{12}^{opt}$; **d** The injected relative power; **e** The relative phase (with respect to a signal injected from TL $\alpha = 1$) of the signal injected from TL $\alpha = 3$. **f** Network schematics for Coherent Perfect Absorption (CPA) for a wave injected from TLs $\alpha = 1, 2$ at frequency $f = 3.26$ GHz. **g** Convergence of the CPA objective function $g_{CPA}$ toward nearly perfect absorption ($g_{CPA} \approx 0.9998$). Evolution of: **h** The electrical length $L_{12}^{opt}$; **i** The injected relative power; **j** The relative phase (with respect to a signal injected from

TL $\alpha = 1$) of the signal injected from TL $\alpha = 2$ vs. iteration number. **k** Network schematics for signal invisibility (cavity-camouflage). The interrogating signal at frequency $f = 0.74$ GHz is injected into the network from TL $\beta_0 = 1$ and is received from TL $\alpha_0 = 3$ with the same amplitude and phase (0.01 dB power variation and $0.1^o$ phase variation with respect to the injected wave). A control signal (phase and amplitude) injected from lead $\alpha_c = 2$ is balancing the losses and together with the length $L_{12}^{opt}$ ensures the invisibility of the cavity as far as the processing signal at TL $\alpha_0 = 3$ is concerned. The reflected signal from lead $\beta_0 = 1$ is essentially zero. **l** Convergence of $g_{invis}$ to $-10^{-4}$, signifying that the transmitted field matches the desired (injected) wave. Evolution of: **m** The cable-length $L_{12}^{opt}$; **n** The injected relative power; **o** The relative phase (with respect to the signal injected from lead $\beta_0 = 1$) of the control signal injected from lead $\alpha_c = 2$ vs. iteration number. The black solid (colored dashed) lines with filled black circles (colored squares) are the experimental (digital twin) results.

NLOPT with the Limited-memory Method of Moving Asymptotes (LD_MMA) option[57]. This algorithm identifies a new set of parameters, which were then implemented by adjusting the phase shifter length and modifying the relative amplitude and phase of the input signals; *(e)* Iteration and Convergence: The steps (a–d) are repeated until the objective function converges on an optimal value (within some tolerance). A single operating frequency was selected and held constant throughout the optimization process.

In Fig. 2 we report the results of the in-situ optimization (solid black lines with filled circles) for the three modalities discussed above. In all cases, we have achieved a rapid convergence towards an optimal value of the corresponding objective functions occurring after ~20 iterations of the protocol, see Fig. 2b,g,l. The convergence of the three control parameters ($L_{12}^{opt}, A, \theta$) towards their optimal value for each of the three modalities is reported in the third, fourth, and fifth rows of the same figure respectively.

### In-silico Implementation of iPAC for large control parameter system

In Fig. 3, we also present the in-silico results (shown as dashed lines with filled squares) obtained from a digital twin implementation of the AO protocol. The close agreement between the digital twin and the in-situ results confirms that our experimental set-up is adequately

captured by our network model. This ad hoc validation supports the applicability of the digital twin approach to more complex networks, with a larger number of control parameters. Needless to stress that, in general/typical cases, an in-silico implementation of the AO is subject to a number of drawbacks like developing a highly accurate model of every propagation path, boundary and loss channel in continuous space which can become computationally burdensome and sensitive to modeling errors, especially for complex cavities. These have to be contrasted with the in-situ implementation of iPAC, where the physical system itself computes its own Green's function in real time, by performing only local measurements.

We considered fully connected networks consisting of $N = 20$ vertices with a total of 190 bonds. The bond-lengths are initially uniformly distributed in the interval $[L_0 - \delta L, L_0 + \delta L]$ where $L_0 = 25$ cm and $\delta L = 5$ cm. At each vertex, we have attached TLs (i.e. $N = 20$) that have been used for injecting (receiving) the interrogating (scattering) signal. The frequency of the injected waves was chosen to be $f = 3.2$ GHz for all cases. The optimization process has been achieved via bond-length variations (cavity-shaping approach).

The first column of Fig. 3 reports the results of the TMT modality. A random wavefront has been injected into the network from six TLs coupled to vertices $n = 1, 3, 6, 7, 9$, and 15. The adjoint optimization scheme aimed to identify the appropriate bond-length variations that

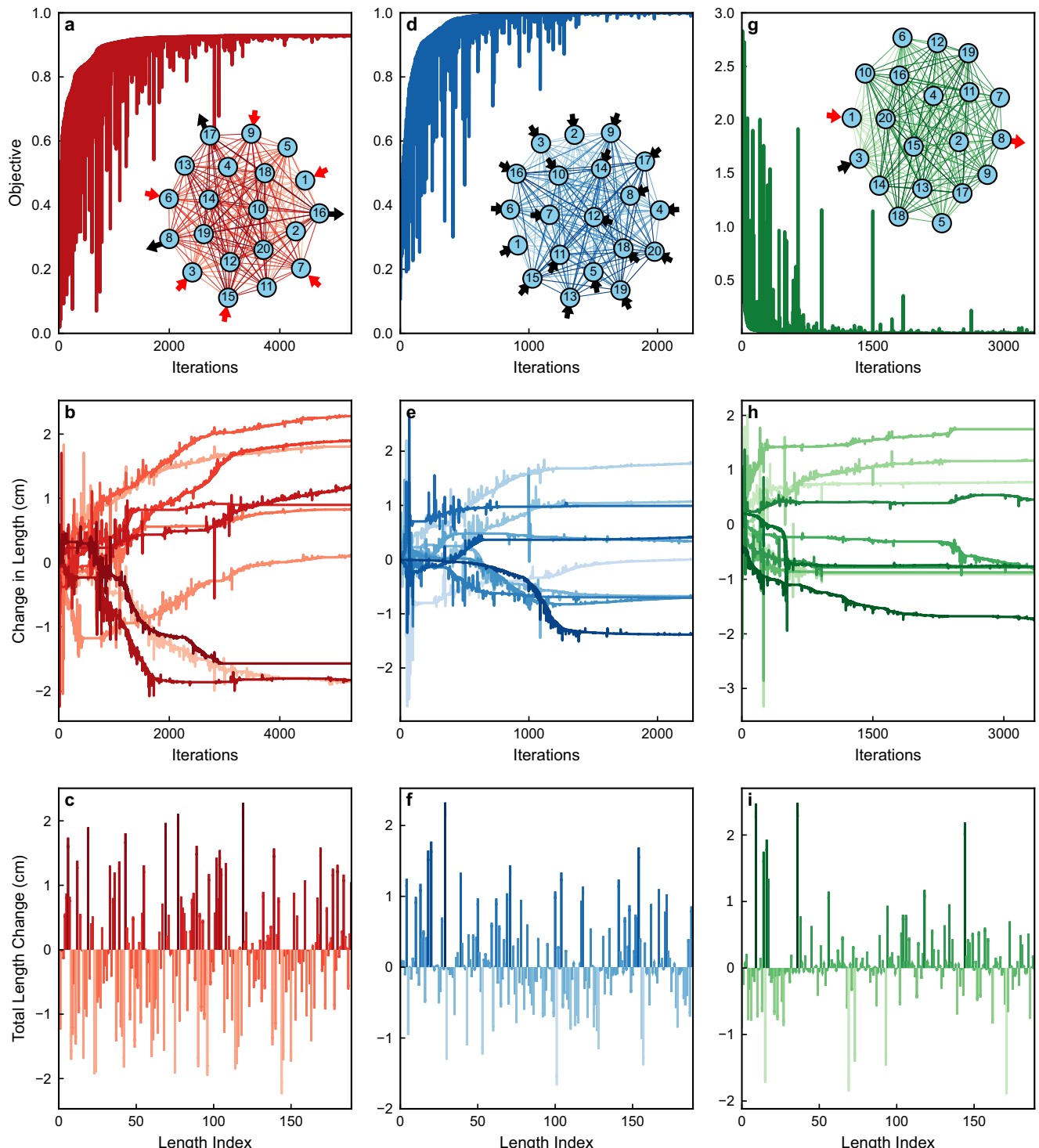

**Fig. 3 | In-silico demonstration of the iPAC optimizer.** In-silico demonstration of the iPAC scheme using a digital twin of a fully connected network of $V = 20$ vertices consisting of 190 lossy coaxial cables. Each vertex is attached to a TL. The control parameters used for the in-silico optimization involve only the bonds of the network (cavity shaping). Three different modalities, all at $f = 3.2$ GHz, are demonstrated: **a–c**: Targeted Mode Transmission (TMT) for a scenario where six TLs $\alpha = 1, 3, 6, 7, 9,$ and 15 are used to inject a random wavefront, and the network is optimized to deliver the input signal $\approx 90\%$ to the targeted channels $n = 8, 16, 17$. **a** Convergence of the TMT objective function $g_{\text{TMT}}$ vs. iteration number. **b** Representative evolution of selected bond-length variations during the optimization. **c** Final set of bond-length variations across all network bonds. **d–f**: Coherent

Perfect Absorption (CPA) scenario for a wave injected from all $N = 20$ TLs. **d** Convergence of the CPA objective function $g_{\text{CPA}}$ toward nearly perfect absorption ($g_{\text{CPA}} \approx 0.9998$). **e** Representative bond-length variations vs. iteration. **f** Final network configuration achieving the CPA state. **g–i**: Signal invisibility (cavity-camouflage). The interrogating signal is injected into the network from channel $\beta_0 = 1$ and is received from TL $\alpha_0 = 8$ with the same amplitude and phase. A control signal injected from a control channel $\alpha_c = 3$ is balancing the losses. **g** Convergence of $g_{\text{trans}}$ to $-6 \times 10^{-4}$, signifying that the transmitted field matches the desired (injected) wave. **h** Evolution of selected bond-length variations during the optimization. **i** Final distribution of bond-length variations across all bonds after 3351 iterations.

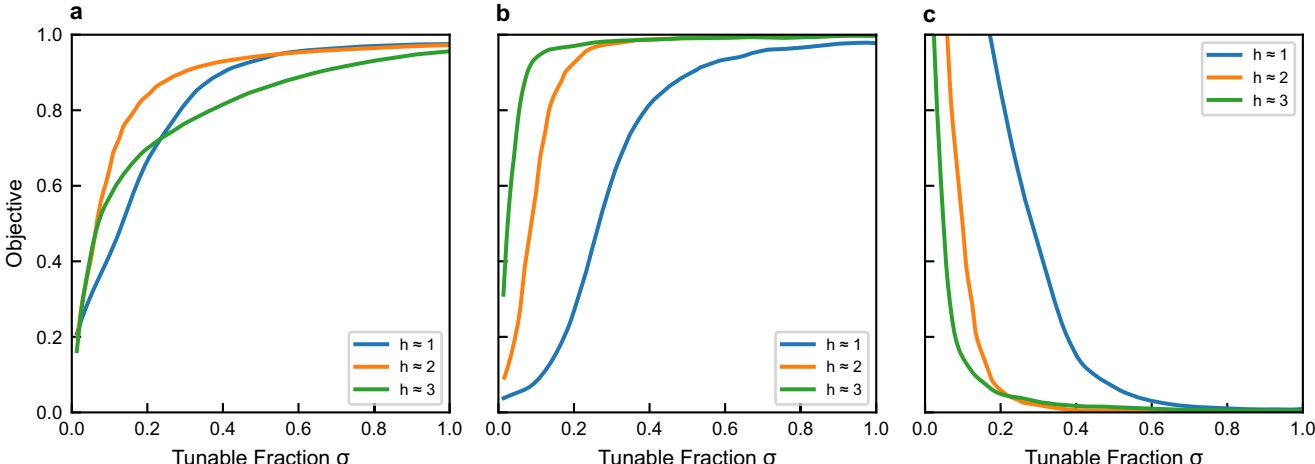

**Fig. 4 | Efficiency of Adjoint Optimization.** Efficiency of the Adjoint Optimization as a function of the number of controlled parameters and network complexity $h_{top}$ for: **a** TMT modality; **b** CPA modality; and (**c**) signal invisibility. Solid lines correspond to three representative networks with $h_{top} \approx 1$ (blue), 2 (orange), 3 (green). Each network has $V = 75$ vertices with $V_{bulk} = 50$ bulk (interior) vertices whose connectivity is varied to tune $h_{top}$, and $V_{TL} = 25$ boundary vertices used to attach transmission lines (TLs). Each TL vertex connects to the bulk via two bonds (valency $v_{TL} = 2$), which keeps the TL-network coupling constant fixed as the bulk connectivity changes (see Methods). In all cases, performance improves as the fraction $\sigma$ of tunable bulk-bulk bonds (controlled parameters) increases. For $h \approx 1, 2, 3$ the total number of bulk bonds are 73, 193, 529, respectively.

resulted in delivering the injected signal to a specified set of channels attached to vertices $n = 8, 16, 17$, see inset of Fig. 3a. In the main part of subfigure Fig. 3a, we show the convergence of the objective function $g_{TMT}$ to a total transmittance of approx. 90%. We have checked that for the converged optimal bond-length configuration, the remaining 10% energy loss was associated with the absorption due to Ohmic losses at the wires. Typical bond-length variations $\delta L_{nm}^{opt}$ versus the iteration number are shown in Fig. 3b, while in Fig. 3c, we report the bond-length variations for all bonds of the network at the end of the optimization process.

The second column of Fig. 3 reports the digital twin calculations for the CPA-scenario. We have injected a randomly chosen coherent wavefront from all $N = 20$ TLs into the lossy network, see inset of Fig. 3d. Using the adjoint optimization protocol we have determined the optimal bond-lengths for which the network acts as a perfect constructive interference trap leading to complete absorption of the incident wave. In Fig. 3d, we show the convergence of the objective function towards a value $g_{CPA} = 0.9998$ occurring after 2276 iterations. The evolution of some typical bond variations versus the number of iterations is shown in Fig. 3e, while Fig. 3f reports the final bond-variation for all bonds.

Finally, the last column of Fig. 3 shows the digital twin simulations for cavity camouflage (invisibility), see the inset of Fig. 3g. We have injected a signal from channel $\beta_0 = 1$ with an amplitude $A_{\beta_0} = 0.56$ and phase $\theta_{\beta_0} = 69^o$. To balance the network losses we have also injected a control signal into the system from channel $\alpha_c = 3$ with amplitude $A_{\alpha_c} = 0.85$ and phase $\theta_{\alpha_c} = 333^o$. The adjoint optimization protocol aimed to identify an appropriate bond-length configuration for which the scattered signal collected at a specified $\alpha_0 = 8$ TL is identical to the interrogating wave injected from TL $\beta_0 = 1$. In Fig. 3g, we show the convergence of the objective function $g_{trans}$ towards the value $g_{invis} \approx 6 \times 10^{-4}$ after 3351 iterations. Here, in addition to the constraints imposed by the objective function $g_{invis}$ in Eq. (9) we have also requested zero transmission and reflection from the control channel $\alpha_c = 3$. This additional constraint introduce the following modification $g_{invis} \rightarrow g_{invis} + \frac{|\Phi_c - A_{\alpha_c} e^{i\theta_{\alpha_c}}|^2}{\sum_{(I_\beta)} |A_\beta|^2}$. Even with a modest number of iterations ~400 of the adjoint optimization scheme the objective function can be

as small as $g_{trans} \approx 0.01$. Figure 3h,i show a representative evolution of bond-length variations versus iteration number and the final bond-lengths at the end of the optimization process (3351 iterations).

## Efficiency of adjoint optimization computing

It is instructive to examine how the efficiency of the Adjoint Optimization scheme varies with the number of controlled parameters **p** and the complexity of the network. The latter can be quantified by the topological entropy $h_{top} = \ln \rho$[58], where $\rho$ is the spectral radius of the non-backtracking (Hashimoto) matrix **T** indexed by the set of directed bonds of the network (see "Methods"). The number of period-$n$ non-backtracking closed paths (counted up to a cyclic shifts) is $\frac{1}{n} \text{Tr}(\mathbf{T}^n) \sim \rho^n/n$, where $\rho$ is bounded between the minimum and maximum valency $v_n$ as $\min_n(v_n - 1) \le \rho \le \max_n(v_n - 1)$. Hence, $h_{top}$ characterizes the combinatorial branching complexity of the underlying non-backtracking dynamics and thus reflects the connectivity of the network.

In Fig. 4 we report the efficiency of the Adjoint Optimization for each of the three modalities as a function of the number of control parameters, for three representative values of the topological entropy. In all cases, we find that the efficiency increases with the number of control parameters. The convergence rate of the objective function to its optimum, however, depends on the task and on $h_{top}$. For CPA, convergence accelerates as $h_{top}$ increases. This finding is consistent with higher connectivity (more lossy bonds), which strengthens multiple scattering, traps energy, and enhances absorption. In contrast, TMT and invisibility show a non-monotonic dependence on $h_{top}$, with fastest convergence near $h \approx 2$ indicating that moderate complexity aids optimization, whereas excessive connectivity slows the approach to the optimal values of the objective functions. Further simulation details are given in Methods.

We stress that gradient-based schemes are far more time-efficient than derivative-free alternatives (e.g., genetic algorithms or particle-swarm methods) once the number of optimization parameters grows large[59]. In high-dimensional settings, non-gradient approaches typically incur substantially greater computational effort and memory, with scaling that deteriorates as the parameter size increases[60,61]. Empirically, gradient methods also frequently attain better local optima[25,59,62,63]. As a representative case,[59] benchmarks Adjoint

optimization against a genetic algorithm for photonic inverse design and finds the former markedly less demanding in computational resources.

Other gradient-based methods that approximate the gradients via finite differences involve solving or simulating the problem described by Eqs. (3),(5) for every combination of the optimization variables $\mathbf{p} = (p_1, p_2, \cdots)^T$ leading to too-long computational times. Instead, the Adjoint Optimization bypasses this costly (in terms of computational memory and time) bottleneck when number of optimization parameters increases, because all the gradients can be computed at once by solving only two distinct problems associated with the forward and the backward propagation. A detail comparison between finite difference and Adjoint Optimization approaches is given in the Supplementary Information.

Of course, the in-silico implementation of an Adjoint Optimization is naturally limited by the computational complexity of simulating the physical system. On the contrary, the proposed in-situ implementation of the Adjoint Optimization which bypasses all the latencies (memory and time complexity) of digital simulations. All other computational steps have nearly fixed cost that do not scale (or scale very weakly) with the system size.

These unique features of the Physical Adjoint Optimization and its in-situ implementation are very promising for potential applications in RISs, where a large number of tunable elements may need to be reconfigured quickly to achieve predetermined, yet changing, goals.

## Discussion

In conclusion, we have presented a proof-of-concept experimental demonstration of in-situ Physical Adjoint Computing for real-time control of electromagnetic wave modalities in complex multi-resonant and multi-scattering systems. Our approach can be mapped onto indoor wireless communication protocols[64], in which a reconfigurable intelligent surface can be swiftly programmed to deliver stronger signals to moving targets amidst an evolving environment in real time. In such protocols, electric field measurements need to be made only at the positions of the metasurface elements and the target. Of course, such reconfigurable intelligent metasurfaces (RIS) will require the integration of sensing capabilities by appropriate modification of the tunable meta-atoms constituting the metasurface. While majority of current RIS are not designed to perform such local measurements, there are some recent works that push the boundaries further proposing[65,66] and even demonstrating[67,68] pixel-level (i.e. per-meta-atom) RF magnitude/phase sensing for closed-loop RIS control.

Importantly, no knowledge is required of the full electromagnetic environment, including any big or small obstacle which may stand in the way or even moving. Therefore, our approach is fundamentally different from existing wavefront-shaping methodologies, which require a full knowledge of the scattering matrix and its eigen-decomposition to identify the optimal wavefront patterns for achieving specific operations. Crucially, in such methods, the entire scattering matrix needs to be repeatedly re-measured and re-analyzed every time the metasurface is reconfigured and/or the surrounding changes, leading to formidable challenges in larger and more complex environments. In contrast, in-situ adjoint optimization bypasses the need for a scattering matrix by directly exploiting the gradient sensitivities judiciously plucked from a set of strategically positioned measurements.

A potential drawback arises in time-varying electromagnetic environments. Three characteristic time scales are relevant: (a) Fast environment vs settling: $\tau_{env} < \tau_{ss}$, the environmental fluctuation (coherence) time $\tau_{env}$ is shorter than the wave settling (steady-state) time $\tau_{ss}$. This regime is atypical and ill-posed for any steady-state optimizer. (b) Quasi-static environment: $\tau_{env} > \tau_{adj}$, the environment changes slower than the convergence time of the in-situ adjoint optimization $\tau_{adj}$. This quasi-static case is ideal for our scheme. (c)

Iteration-timescale drift: $\tau_{env} \propto O(\tau_{it})$, the environmental fluctuations are on the order of the per-iteration time $\tau_{it}$. We analyze this regime in the Supplementary Information under the worst-case condition $\tau_{env} = \tau_{it}$. There, we assume bond-parameter variations on the timescale of a single iteration and evaluate the optimization efficiency versus the variation strength for each of the three modalities reported in the main text. The results are task-dependent and show differing robustness to environmental fluctuations. In our experiment, the iteration time is $\approx 1$ s, with the bottleneck set by relatively slow mechanical phase shifters. Faster actuation (e.g., semiconductor-based) would enable regimes with $\tau_{env} > \tau_{it}$, yielding improved performance under fluctuations.

Let us finally point out that our wave-network platform also significantly differs from physical implementations of feed-forward neural networks, which typically do not utilize complex (multi-scattering) wave interactions in a multi-resonant electromagnetic environment. By leveraging these interactions, our platform amplifies small variations in the optimization parameters via multiple interference pathways, deriving richer physical abilities from a relatively smaller number of controllable parameters (in contrast to billions of weights and biases required in a feed-forward neural net). While we do not pursue any deep learning functionality in this work (and thus require no training data), we note that our setup offers a natural "physics-aware" deep learning platform for both in-situ training and inference, rather than a cumbersome imitation of an abstract neural network architecture. Most importantly, our experiments pave the way for the development of more powerful in-situ optimization protocols which will involve nonlinear and non-reciprocal wave mechanics, broadband pulses, and real-time control learning.

We conclude our discussion by pointing out that the implementation of the in-situ Physical Adjoint Computing can be extended beyond in-doors wireless communications. The same protocol can be applied to a variety of other wave-physics frameworks, including adaptive optics (e.g. for turbulence compensation, ophthalmic imaging, deep-tissue microscopy, multimode-fiber endoscopy), seismic wave control etc.

## Methods
### Network modeling
The transport properties of the microwave network are modeled using a metric graph consisting of one-dimensional wires (bonds) supporting a single propagating mode. The waves propagating between the inner and the outer conductor along the coax cable, is given in terms of the difference $\psi_B(x_B)$ between the potentials at the conductors' surfaces, see Eq. (4). The bonds are connected together at vertices ($v$-port dividers) where Neumann boundary conditions are imposed. In the experimental network we have used $3-port$ vertices (T-junctions).

The solutions of Eq. (4) at a bond $B = (nm)$ can be expressed as

$$\psi_B(x_B) = \phi_n \frac{\sin[k(L_B - x_B)]}{\sin(kL_B)} + \phi_m \frac{\sin(kx_B)}{\sin(kL_B)} \qquad (10)$$

which satisfy the wave continuity conditions $\psi_B(x_B = 0) = \phi_n$; $\psi_B(x_B = L_B) = \phi_m$, for each pair of connected vertices $n < m$. Furthermore, the current is conserved at each vertex, i.e.,

$$\sum_m \frac{d\psi_B(x_B)}{dx_B}\Big|_{x_B = 0} + \sum_{\alpha'=1}^{L} \delta_{\alpha\alpha'} \frac{d\psi_{\alpha'}(x)}{dx}\Big|_{x=0} = 0, \qquad (11)$$

where $\delta_{\alpha\alpha'}$ is the Kronecker delta, and the second term accounts for the derivatives at the ports connected to TLs. Combining the above vertex boundary conditions, together with Eq. (10) we arrive at Eq. (5) which provides the system matrix that describes transport in the forward direction.

## Characterization of network complexity

We quantify the network complexity by the topological entropy $h_{\text{top}} = \ln \rho$ where $\rho$ is the spectral radius of the $2B \times 2B$ non-backtracking (Hashimoto) matrix $\mathbf{T}$. The latter is defined in the space of directed edges of the network $b = (n \to m)$ with entries

$$T_{b,b'} = 1, \text{ if } m(b') = n(b) \text{ and } b \neq \bar{b}' \text{ (no immediate backtracking)}$$
$$= 0, \text{ otherwise} \tag{12}$$

with $\bar{b} \equiv (m \to n)$ being the reverse arc. Notice that for any row $b$ the sum $\sum_{b'} T_{bb'} = v_{n(b)} - 1$ where $v_{n(b)}$ is the valency of the vertex $n$. Therefore, $\min(v_{n(b)} - 1) \leq \rho \leq \max(v_{n(b)} - 1)$.

Since $\mathrm{Tr}\mathbf{T}^n$ counts the number of period-$n$ non-backtracking closed paths on the network,

$$h_{\text{top}} = \ln \rho = \lim_{n \to \infty} \frac{1}{n} \ln \mathrm{Tr}\mathbf{T}^n. \tag{13}$$

More branching and more loops indicate more routes and consequently larger $h_{\text{top}}$; sparse or weakly looped graphs yield small $h_{\text{top}}$ (for intuition, a $v$-regular network with $v_n = v$ for all $n$ has topological entropy $h_{\text{top}} = \ln(v - 1)$ whereas a single cycle network has $h_{\text{top}} = 0$). Thus $h_{\text{top}}$ is a pure connectivity metric, which is relabeling-invariant, independent of bond lengths, that captures the combinatorial growth of admissible routes on the network.

## In-silico Implementation of the iPAC

We conducted in-silico simulations of a complex scattering network consisting of $V = 20$ vertices that are fully connected with 190 bonds. Each vertex was attached to a TL, i.e., $N = 20$. In the digital twin simulations, we considered a cavity-shaping optimization scheme that allows adjustments of all 190 bond-length, i.e., a large number of DoF relevant to operational realities. In fact, under such conditions is expected that the implementation of the adjoint-based gradient descent protocol is more beneficial as compared to other optimization schemes

The bond lengths were initialized with randomized values uniformly distributed around $L_0 = 0.25$ m, with variations constrained to $\delta L \pm 5$ cm ($= 0.5\lambda$ where $\lambda$ is the operational wavelength), such that $L_{nm} \in [L_0 - \delta L, L_0 + \delta L]$. For all modalities, the wavefront parameters were fixed, with the amplitude parameters constrained $A_i \in [0.001, 3.0]$, and $\theta_i \in [-\pi, \pi]$. An amplitude lower bound was set to prevent trivial solutions with zero input power.

## Controlling and setting up the VNA for coherent wavefront inputs

To precisely control the phase and amplitude of signals injected into the input ports of the scattering system, a Keysight PNA P5023B four-port Vector Network Analyzer (VNA) equipped with the S93089B Differential and I/Q Device Measurements option was utilized. The S93089B option enables accurate phase control of multiple internal sources, facilitating coherent excitation without the need for external hybrid couplers or baluns. Two internal sources were configured to deliver signals to the desired input ports of our microwave graph network. Both sources were set to the same frequency to maintain coherence, while the relative phase between the two sources was precisely adjusted from $0^o$ to $360^o$ using the S93089B's phase control settings. This allows the phase difference to be fixed at specific values in degrees of one input port relative to a reference port. The output power of each source was individually adjusted in the interval $[-40 \text{ dBm}, 0 \text{ dBm}]$ to achieve the desired amplitude difference at each input port. A calibration routine was executed to compensate for any inherent phase and amplitude imbalances introduced by the VNA's internal signal paths and external cabling. For ensuring the experimental stability of the objective function, we performed ten measurements per iteration, resulting in essentially identical outputs characterized by their mean value.

For forward scattering measurements, both sources were activated and phase-aligned according to a random set of initial values. The S93089B option's source-phase control ensured that the relative phase between the inputs was maintained with high precision throughout the measurement. The VNA's receivers were configured to measure the DUT's response at the fundamental frequency, capturing the effects of coherent excitation on the forward scattering parameters. In adjoint measurement scenarios, these sources provided the excitation signal, and the VNA measured the reflected and transmitted signals accordingly (in case only one excitation signal was needed the other source was deactivated).

## Controlling and setting up the mechanical phase shifter

To achieve precise length manipulation in our experimental setup, we integrated a mechanical phase shifter into the system (bond $L_{12}^{\text{opt}}$), a coaxial RF phase shifter, typically operated via a manual knob. To enable automated and repeatable control, we motorized the phase shifter and developed a characterization method to correlate motor movements with the resulting length perturbations.

The mechanical phase shifter used in our experiment was designed to operate over a frequency range of DC to 18 GHz, with an insertion loss of less than 1.0 dB up to 18 GHz and capable of handling up to 100 Watts of average RF power. The device originally featured a manual adjustment knob for phase tuning, however, a stepper motor was mechanically coupled to the phase shifter's adjustment knob. The motor was securely mounted to maintain alignment and prevent mechanical backlash, ensuring consistent control over the phase adjustment mechanism. The motor was interfaced with a Trinamic motion controller, allowing for precise digital control of the motor's position, and providing an adjustable phase shifter's length from 3 mm to 23 mm. The controller was connected to a computer via a USB interface, enabling automated control through custom Python scripts.

To establish a reliable relationship between the motor's rotational steps and the physical displacement within the phase shifter, we conducted a calibration process using a high-precision digital micrometer (Asimeto IP65 Digital Outside Micrometer). The micrometer was positioned to measure the linear displacement resulting from the motor's rotation. The micrometer's spindle was placed in contact with a reference point on the phase shifter that moved in response to the internal adjustment mechanism. The motor was programmed to move in increments of microsteps, and the corresponding displacement was recorded using the micrometer. Movements were performed in both clockwise and counterclockwise directions to account for any mechanical hysteresis. The collected data indicated that 464 microsteps of the motor corresponded to a linear displacement of 1 mm within the phase shifter. A linear relationship was established between the number of micro-steps ($N_{steps}$) and the displacement $d = \frac{N_{steps}}{464} m$mm. Multiple trials were conducted to confirm the repeatability of the calibration. The standard deviation of the displacement measurements was within the micrometer's specified accuracy, ensuring confidence in the calibration.

Finally, we point out that the accuracy of the phase shifter was depending on the resolution of our micrometer, which is roughly 0.01 mm. However, the optimizer used to update the optimization parameters, treated this length as a continuous, real-valued set-point, with higher precision than it is measured. Despite this "quantization" error, there was no issue with our optimization performance as the gradients pointed in the correct direction. In fact, a degree of stochasticity in the gradients are known to be beneficial for gradient-guided optimization, as our in-situ studies confirmed due to an unavoidable stochasticity whose origin is traced back to various experimental noise sources (temperature variations, technical noise at the equipments etc).

## Modeling of the phase shifter

To understand the phase shifter's impact on the transmitted signals, we modeled it as a variable-length transmission line supporting a Transverse Electromagnetic (TEM) mode. The phase shift introduced by the device is a function of the electrical length, which depends on both the physical length and the dielectric properties of the medium. Using the calibrated Vector Network Analyzer (VNA) setup described previously, we measured the scattering matrix (S-parameters) of the phase shifter over the frequency range of interest. Measurements were taken at various positions of the phase shifter corresponding to different micrometer readings. The phase shifter was modeled as a two-port network with its behavior represented by transmission line equations. The phase shift ($\phi$) introduced by the line is given by $\phi = \beta \cdot d$ where $\beta$ is the phase constant, and $d$ is the physical length of the transmission line. The phase constant is related to the frequency $f$ and the effective permittivity $\epsilon_{eff}$ of the medium by $\beta = \frac{2\pi f}{c}\sqrt{\epsilon_{eff}}$ where $c$ is the speed of light in a vacuum. The effective permittivity was assumed to be complex to account for dielectric losses within the phase shifter. We modeled $\epsilon_{eff}$ as a function of the micrometer-measured length and frequency. To extract the relationship between the effective permittivity, physical displacement, and frequency, we employed surrogate optimization using MATLAB, finding that $n = \epsilon_{eff} \approx 1.004 + 0.0022i$ and the functional dependence of the length of the phase shifter $L_S$ on the measured length of the micrometer $L_{ps} = 286mm + 2(d - 7mm)$ accounting for the fact that it is a trombone line phase shifter, so the factor of 2 accounts for the doubling of the line when making length adjustments.

## Efficiency of adjoint optimization methodology

### Efficiency for increasing complexity and optimization parameters.

To assess how combinatorial complexity and tunability affect iPAC efficiency, we simulated three modalities (TMT, CPA, invisibility) across three topological-entropy levels $h_{top} = 1, 2, 3$ (Fig. 4a–c). For each $h_{top}$, we averaged the optimal objective over 100 random networks with identical connectivity (thus the same $h_{top}$) but uniformly random bond lengths $L_b \in [21.2, 28.8]$ cm. Each network had $V = 75$ vertices: a bulk of $V_{bulk} = 50$ and $V_{TL} = 25$ boundary vertices used to attach transmission lines (TLs). We increased $h_{top}$ by raising the bulk connectivity (vertex valencies). TL vertices were kept at valency $v_{TL} = 2$, fixing the TL-network coupling $\Gamma = 1 - |\langle S_{\alpha,\alpha'}\rangle|^2 = 1 - \left(1 - \frac{2}{v_{TL}+1}\right)^2$[44] across realizations and different $h_{top}$ ($S_{\alpha,\alpha'}$ are the diagonal elements of the scattering matrix and $\langle \cdots \rangle$ indicates an averaging over frequency realizations and channels $\alpha$).

The networks with $h_{top} = 1, 2, 3$, involve 73, 193, and 529, respectively, number of bonds that connect the bulk vertices. In our simulations, we varied the number of control parameters by varying the tunable fraction $\sigma$ of bulk-bulk bonds ($\sigma = 1$ indicates that all bulk bonds are tuned). Tunable bond-lengths were constrained within the range $17.3 \le L_b \le 33.7$ cm. The optimization used NLopt's MMA (gradient-based) with relative tolerances $10^{-7}$. For each $(h, \sigma)$ we report the mean optimum over the 100 network realizations. The optimization scenarios used in this analysis were the same ones discussed in the main text. Specifically: (i) for the TMT we have injected a random monochromatic wavefront from 10 TLs and targeted other 5 TLs; (ii) for the CPA, we have injected a random monochromatic wavefront from all 25 TLs; and (iii) for the invisibility we have used one interrogation TL from where the incident signal was injected, and one receiving TL where we have requested the emitted signal to have equal amplitude and phase as the injected one. Finally, we have also employ a separate control TL to balance losses (as in the main text).

Our analysis show that the optimized objective increases monotonically with $\sigma$, and high performance shifts toward $\sigma \to 1$ for all $h_{top}$. The convergence behavior depends on the task and on $h_{top}$: for CPA, convergence accelerates as $h$ increases-consistent with more lossy bonds (from higher connectivity) and stronger multiple scattering that

trap energy and enhance absorption. In contrast, TMT and invisibility show a non-monotonic dependence on $h_{top}$, with fastest convergence near $h_{top} \approx 2$; moderate complexity aids optimization, whereas excessive connectivity offers diminishing returns. A detailed analysis of this non-monotonicity is left for future work.

**Time-performance benchmark.** We model four latencies: measurement $t_m$, interconnect $t_l$, host compute $t_c$, and actuation $t_o$. One adjoint iteration (forward + adjoint measurement, gradient assembly, update) yields

$$t_{adj}(p) = 2t_m + 6t_l + (2 + p)t_c + \alpha t_0$$

where $p$ is the number of tunable parameters and $\alpha \ge 1$ is the actuation multiplier per update. The constants conservatively account for round-trip link events in the two measurements and a single actuation block.

A central finite-difference (FD) iteration perturbs each parameter $n$ by $\pm\delta$, measures twice, restores the nominal value, and then updates, giving:

$$t_{FD}(p) = t_c + t_m + \alpha t_0 + 4t_l + p(t_c + 2t_m + 3t_o + 10t_l).$$

On our setup (Keysight P5023B VNA; Intel Core i9-14900K; stepper-motor actuators) $t_m = 6.7$ ms; USB microframe: $t_l = 125\,\mu s$; host compute: $t_c = 3$ ns; stepper actuation: $t_o = 0.1$ s per micro-move with $\alpha \approx 10$ resulting in $\alpha t_o \approx 1$s. Thus $t_{adj}(p) \approx 1.0006$ s (for $p \le 10^3$) while FD grows linearly with slope $10t_l + 2t_m + 3t_o - t_c \approx 0.301$ s/parameter, giving $t_{FD}(p = 1000) \approx 302$ s. For further details see Supplementary Information and associated Supplementary Fig. 1.

We, therefore, conclude that since the Adjoint uses two measurements and one update per iteration, it results to a runtime which is essentially independent of $p$. Instead, FD adds two measurements and $\approx 3$ actuations per parameter, making runtime linear in $p$ and dominated by actuation/I/O.

## Data availability

The datasets generated during and/or analyzed during the current study are available in the Zenodo repository[69] (https://doi.org/10.5281/zenodo.17314682).

## Code availability

The algorithms used for this study are standard and are outlined in the Main text and in "Methods." The corresponding authors can provide code scripts upon request.

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

## Acknowledgements

T.K. acknowledges useful discussions with S. Johnson on adjoint methodologies during a Simons Foundation webinar. We acknowledge partial funding from MURI ONR-N000142412548 (J.G. and T.K.), DOE DE-SC0024223 (J.G., C.W., Z.L., and T.K.), NSF-RINGS ECCS 2148318 (J.G., C.W., and T.K.), Simons Foundation SFI-MPS-EWP-00008530-08 (J.G., C.W., and T.K.), and ARO W911NF2410390 (Z.L.).

## Author contributions

J.G. performed the experiments and analyzed the measurements. J.G. and C.W. performed the in-silico calculations. C.W. developed the A.O. protocol under the supervision of Z.L. J.G. performed the time-performance benchmark, complexity-performance analysis and noise analysis. T.K. conceived and initiated the project. Z.L. and T.K. wrote the paper with input from J.G. and C.W. T.K. supervised the execution of the project.

## Competing interests

The authors declare no competing interests.
