## [Transparent Peer Review file · Nature Communications]

In-situ Physical Adjoint Computing in multiple-scattering electromagnetic environments for wave control

Corresponding Author: Professor Tsampikos Kottos

Version 0:

Reviewer comments:

Reviewer #1

(Remarks to the Author)

In this manuscript, the authors achieved in-situ manipulation of the multiple scattering of electromagnetic waves using time- and energy-efficient adjoint optimization methodologies, and demonstrated real-time wave-driven functionalities such as targeted channel emission, coherent perfect absorption, and camouflage.

Overall, both experimental and numerical results are technically sound and impressive. Three examples of targeted channel emission, coherent perfect absorption, and camouflage illustrate the power of in-situ physical adjoint optimization for real-time control of electromagnetic wave modalities in complex scattering systems. One important advantage of the in-situ adjoint optimization is that the full knowledge of the electromagnetic environment is not needed. This is fundamentally different from the wavefront shaping techniques that require repeated measurement of the scattering matrix when the scattering system is reconfigured.

I have a few questions and comments for the authors.

1. The complex scattering environment that the authors utilized can amplify small variations in the optimization parameters via multiple interference pathways. This requires controlling these parameters with high accuracy. What accuracy is needed to vary the coaxial cable length using the phase shifter in the microwave experiment? What happens if such accuracy is not reached?
2. In the experimental results shown in Fig. 2, the authors chose to vary the cable length between vertices 1 and 2. Is there any reason for choosing this particular bond in all three tasks? I wonder what will happen if the authors vary the cable length connecting other vertices? Is it always sufficient to vary only one bond length for different tasks?
3. The numerical simulations were conducted on fully connected networks, but experimental networks were not fully connected. How does the connectivity of a network affects the outcome of physical adjoint optimization? Is a fully connected network the best for physical adjoint optimization? Experimentally it may be difficult to achieve fully connected networks. Is there a minimum number of bonds needed to achieve specific modalities via physical adjoint optimization?

Reviewer #2

(Remarks to the Author)

This manuscript presents an in-situ physical adjoint computing (IPAC) method for wave control in multiple-scattering electromagnetic environment. The proposed IPAC method consists of three components including in-situ measurements, targeted perturbations, and an external control mechanism, to implement a real-time optimization for wave-system. A multi-resonant, multi-scattering complex network is used to mimic the multiple-scattering electromagnetic environment to test the proposed IPAC method. However, there are several concerns in this manuscript as follows.

1. What is the theoretical computational complexity for the proposed IPAC approach? Please provide more analysis for the computational complexity.

2. It is unknown what is the performance of the proposed IPAC approach for large-scale multiple-scattering scenarios. The scale of the multi-resonant multi-scattering complex network is limited. For example, for the example of reconfigurable intelligent surface, the number of scattering elements can be up to 1000. How does the proposed IPAC approach work in such large-scale case?
3. A real example of multiple-scattering electromagnetic environment should be tested to verify the proposed IPAC approach. The multi-resonant multi-scattering complex network is over-simplified. For real multiple-scattering electromagnetic environment, it can be time-varying with a short coherent time. Thus, it is questionable whether the proposed method can work in such real environment.
4. For the in-situ measurement component in the proposed IPAC approach, the measurement will inevitably have errors. Therefore, what is the impact of the measurement error to the proposed approach? How robust is the proposed IPAC approach?
5. For some examples of multiple-scattering electromagnetic environment, such as reconfigurable intelligent surface, the objective function cannot be measured locally. For example, the objective of RIS can be optimizing the channel capacity at the receiver which is far-away from the RIS. Therefore, the proposed approach has some limitations. The authors should discuss this problem.
6. There is no comparison benchmark in the experiment results to verify the benefit of the proposed IPAC method.

Reviewer #3

(Remarks to the Author)

In this manuscript, Guillamon et al. experimentally demonstrate adjoint-based optimization of wave networks with examples including targeted mode transmission, coherent perfect absorption, and invisibility. The adjoint derivation for wave networks, as expressed in Eqs. (5) and (6), appears sound and, in my view, constitutes the primary strength and novelty of this work. Accordingly, I am inclined to recommend this manuscript for acceptance, provided the authors address the following points.

First, regarding the invisibility network in Fig. 2: the schematic does not match the caption. It seems there should be an outgoing wave from lead 2 with amplitude A_1 , while the control signal is injected through lead 3. Additionally, the vertex index for lead 3 should presumably be 3, not 9.

Second, I have some reservations about Fig. 3. I agree with the general idea that the advantage of wave networks lies in reducing complex scattering phenomena in continuous media to a discrete, lumped network of vertices and links. Thus, the demonstration in Fig. 2 with small-size examples is fully convincing. However, while the authors discuss scaling the network up to larger sizes ($N \sim 20$), they fail or do not demonstrate in-situ optimization at this scale, despite claiming in the formalism section (page 3) that their protocol "bypasses" computationally intensive tasks. This ironically raises a question: why not simply optimize in silico if in-situ optimization is even more challenging? I suggest that the authors clarify the practical limitations and potential trade-offs between in-situ and in-silico implementations.

Version 1:

Reviewer comments:

Reviewer #1

(Remarks to the Author)

The authors have addressed all my questions and comments. I recommend the revised manuscript for publication in Nature Communications.

Reviewer #2

(Remarks to the Author)

The authors have addressed most of my concerns.

Reviewer #3

(Remarks to the Author)

I appreciate the authors' clarification on my previous concern. I have no further comments and recommend acceptance of the revised manuscript.

Point-by-point Response to Reviewers Questions for manuscript NCOMMS-25-20992-T

Reviewer #1:

- Overall, both experimental and numerical results are technically sound and impressive. One important advantage of the in-situ adjoint optimization is that the full knowledge of the electromagnetic environment is not needed. This is fundamentally different from the wavefront shaping techniques that require repeated measurement of the scattering matrix when the scattering system is reconfigured.

We thank the reviewer for their positive evaluation of our work and for their thoughtful comments. Please find below a point-by-point response to reviewer's questions/ comments.

- The complex scattering environment that the authors utilized can amplify small variations in the optimization parameters via multiple interference pathways. This requires controlling these parameters with high accuracy. What accuracy is needed to vary the coaxial cable length using the phase shifter in the microwave experiment? What happens if such accuracy is not reached?

The accuracy of the phase shifter depends on the resolution of our micrometer, which is roughly 0.01mm. However, the optimizer used to update the parameters treated this length as a continuous, real-valued set-point, with higher precision than it is measured. Despite this “quantization” error, there was no issue with our optimization performance as the gradients will point in the correct direction. In fact, a degree of stochasticity in the gradients are known to be beneficial for gradient-guided optimization – we have confirmed that this is the case in our experimental studies where I degree of stochasticity, due to various noise sources, is always present. We have added this information at the end of the Methods Section “Controlling and setting up the mechanical phase shifter”.

- In the experimental results shown in Fig. 2, the authors chose to vary the cable length between vertices 1 and 2. Is there any reason for choosing this particular bond in all three tasks? I wonder what will happen if the authors vary the cable length connecting other vertices? Is it always sufficient to vary only one bond length for different tasks?

There is no particular reason for choosing this bond over any other for any of the three tasks, other than it was convenient for the experimental networks of all three modalities that we constructed. We could replace one of the fixed coaxial cables with the phase shifter and still be able to achieve the tasks. The only requirement is to be able to probe the fields at the vertices associated with the cable that incorporates the phase-shifter. For our examples, it turned out that varying one cable-length is sufficient to achieve the targeted modalities. We have added this information at the end of the first paragraph of the Section “In-Situ Implementation of IPAC”.

- The numerical simulations were conducted on fully connected networks, but experimental networks were not fully connected. How does the connectivity of a network affects the outcome of physical adjoint optimization? Is a fully connected network the best for physical

adjoint optimization? Experimentally it may be difficult to achieve fully connected networks. Is there a minimum number of bonds needed to achieve specific modalities via physical adjoint optimization?

The gradient-based adjoint optimization becomes more powerful when the number of optimization parameters (tunable bond-lengths) is larger. Nevertheless, the methodology is also applicable for networks with more limited number of tunable (via phase-shifters) coaxial cables (with potential degradation performance). To address this important point, we have introduced a new section in the main text under the title “Efficiency of Adjoint Optimization Computing”. In this section, we have included additional ablation studies where we have varied the complexity of the network (directly related to the number of connected bonds) and the number of optimization parameters and estimated the performance of the Adjoint Optimization Computing. We have also introduced new Methods sections (“Characterization of Network Complexity” and “Efficiency of Adjoint Optimization Methodology”) which further describe the methodology that we have used in order to qualify the efficiency of the iPAC in terms of increasing complexity (subsection “Performance Efficiency for increasing Complexity and Optimization Parameters” in Methods).

In Methods, we have also incorporated a new paragraph “Time-Performance benchmark” that includes a comparison of time-efficiency between finite difference and Adjoint method for the evaluation of gradients. Further details are given in the supplementary material.

In the Supplementary Material, we have also, incorporated an analysis on the noise-sensitivity where it is assumed that the network is dynamically changing. The conclusions of this analysis are summarized in a new paragraph at the end of the “Discussion” section.

The question of “what is the minimum parameter space” is modality-dependent and could be potentially answered by recent methodologies on convex relaxations and bounding on inverse design. This question is beyond the scope of the present paper and might be pursued in a future work.

Reviewer #2

- This manuscript presents an in-situ physical adjoint computing (IPAC) method for wave control in multiple-scattering electromagnetic environment. The proposed IPAC method consists of three components including in-situ measurements, targeted perturbations, and an external control mechanism, to implement a real-time optimization for wave-system. A multi-resonant, multi-scattering complex network is used to mimic the multiple-scattering electromagnetic environment to test the proposed IPAC method. However, there are several concerns in this manuscript as follows.

We thank the reviewer for their thoughtful comments on our work. Please find below a point-by-point response to reviewer’s questions/comments and concerns.

- What is the theoretical computational complexity for the proposed IPAC approach? Please provide more analysis for the computational complexity.

An in-silico implementation of an adjoint optimization is naturally limited by the computational complexity of simulating the physical system. This is in contrast with the proposed in-situ

implementation of the adjoint optimization which bypasses all the latencies (memory and time complexity) of digital simulations. All other computational steps have nearly fixed cost that do not scale (or scale very weakly) with the system size. For example, increasing the number of optimization parameters do not make the in-situ optimization more costly (in terms of computational memory and time) because all the gradients *can be computed at once* via backward propagation. We have stressed this point at the end of the new section “Efficiency of Adjoint Optimization Computing” that we included in the main text of the revised manuscript.

- It is unknown what is the performance of the proposed IPAC approach for large-scale multiple-scattering scenarios. The scale of the multi-resonant multi-scattering complex network is limited. For example, for the example of reconfigurable intelligent surface, the number of scattering elements can be up to 1000. How does the proposed IPAC approach work in such large-scale case?

We appreciate this question which has motivated us to perform additional in-silico simulations of large scale networks in the revised manuscript, representing even more complex multiple scattering scenarios. We have now introduced a new section “Efficiency of Adjoint Optimization Computing” which addresses this question (including other elements associated with the efficiency of the Adjoint scheme). A new Methods subsection “Time-Performance Benchmark” and additional Supplementary Material are also now included that present comparison of latency of Adjoint Optimization Computing with standard finite different methods.

- A real example of multiple-scattering electromagnetic environment should be tested to verify the proposed IPAC approach. The multi-resonant multi-scattering complex network is oversimplified. For real multiple-scattering electromagnetic environment, it can be time-varying with a short coherent time. Thus, it is questionable whether the proposed method can work in such real environment.

In general, one has to distinguish between three possible time-scales for the environmental change: (a) the environmental fluctuations (coherence) time τ_{env} being shorter than the time $\tau_{steady-state}$ that takes for the coherent electromagnetic waves to settle into a steady-state i.e. $\tau_{env} \leq \tau_{steady-state}$; (b) The environment is changing slower than the time that it takes for the real-time in-situ adjoint optimization $\tau_{adjoint}$ to converge i.e. $\tau_{env} \geq \tau_{adjoint}$; and (c) The environmental fluctuations time is of the order of the time $\tau_{iteration}$ that is needed for a few iterations of the real-time in-situ adjoint optimization i.e. $\tau_{env} \propto O(\tau_{iteration})$. The first case is atypical while the second case is the best possible scenario for our scheme. We have focused on the third case and performed additional investigations where we have assumed the worst-case scenario corresponding to $\tau_{env} = \tau_{iteration}$. Specifically, we have assumed variations of the bond of the network at the time scale of one iteration and analyzed the optimization efficiency in terms of the variation strength of the bonds for each of the three modalities that we exhibit in the main text. The results are task-dependent, showing a varying degree of robustness against the strength of the environmental variations. In our experiment the iteration time is of the order of 1sec with a bottleneck associated to the rather slow mechanical phase-shifters. Obviously, more advanced actuation schemes (e.g. semiconductor-based) will allow better scenarios where $\tau_{env} > \tau_{iteration}$; thus, leading to even better performance amidst fluctuations. We have included such discussion at the Discussion section of the revised manuscript, while a further analysis is presented in the

supplementary material at section “Performance of Adjoint Optimization Method in Time-Varying Environments”.

We want, however, to stress that the main point of our paper is the demonstration of the experimentally feasible concept of real-time gradient based adaptive optimization which typically has the fastest convergence properties among various optimization strategies. Of course, the advantage of the method depending on the circumstances. We envision a variety of controlled quasi-static, or slowly-varying yet multi-scattering environments, e.g. real-time control of tabletop experiments, real-time earthquake management and energy harvesting, or adaptive ranging and geo-satellite imaging, where the applicability of the method is beneficial. We have reinforced this point at the Introduction (see also the new Fig. 1) and at the Discussion sections of the revised manuscript.

- For the in-situ measurement component in the proposed IPAC approach, the measurement will inevitably have errors. Therefore, what is the impact of the measurement error to the proposed approach? How robust is the proposed IPAC approach?

The optimizer used to update the parameters treated them as continuous, real-valued set-points – more precise than what we can measure. Despite this “quantization” error, there was no issue with our optimization performance as the gradients will point in the correct direction. In fact, a degree of stochasticity in the gradients are known to be beneficial for gradient-guided optimization – we have confirmed that this is the case in our experimental studies. We have clarified this point at the end of the Section “Controlling and setting up the mechanical phase shifter” at the Methods section.

- For some examples of multiple-scattering electromagnetic environment, such as reconfigurable intelligent surface, the objective function cannot be measured locally. For example, the objective of RIS can be optimizing the channel capacity at the receiver which is far-away from the RIS. Therefore, the proposed approach has some limitations. The authors should discuss this problem.

We agree that the proposed approach requires the ability to probe the fields at the target/receiver and therefore is not applicable to situations where the receiver is far away and cannot be controlled. We have clarified this statement in the revised manuscript. At the same time, we point out that while majority of current RIS schemes are not designed to perform local measurements (as the reviewer points out), there are some recent works that push the boundaries further proposing (and even demonstrating proof-of-principle) pixel-level (per-meta-atom) RF magnitude/phase sensing for closed-loop RIS control. We have indicated some of these references (see Refs. [68,69,70,71]) and discuss this outlook in the Discussion section (see end of first paragraph) as the reviewer requested.

- There is no comparison benchmark in the experiment results to verify the benefit of the proposed IPAC method.

We have added extra benchmark studies in the main text (see the new section “Efficiency of Adjoint Optimization Computing”) and the Methods (see new section “Time-Performance

Benchmark”) and Supplementary Material (see Section “S1 Time-Performance Benchmark”) which now is included in the revised manuscript.

Reviewer #3

- In this manuscript, Guillaumon et al. experimentally demonstrate adjoint-based optimization of wave networks with examples including targeted mode transmission, coherent perfect absorption, and invisibility. The adjoint derivation for wave networks, as expressed in Eqs. (5) and (6), appears sound and, in my view, constitutes the primary strength and novelty of this work. Accordingly, I am inclined to recommend this manuscript for acceptance, provided the authors address the following points.

We thank the reviewer for their positive evaluation of our work and for their thoughtful comments. Please find below a point-by-point response to reviewer’s questions/ comments.

- First, regarding the invisibility network in Fig. 2: the schematic does not match the caption. It seems there should be an outgoing wave from lead 2 with amplitude A_1 , while the control signal is injected through lead 3. Additionally, the vertex index for lead 3 should presumably be 3, not 9.

Thank you! We have corrected Fig. 2 and the associated portion of the figure caption accordingly.

- Second, I have some reservations about Fig. 3. I agree with the general idea that the advantage of wave networks lies in reducing complex scattering phenomena in continuous media to a discrete, lumped network of vertices and links. Thus, the demonstration in Fig. 2 with small-size examples is fully convincing. However, while the authors discuss scaling the network up to larger sizes ($N \sim 20$), they fail or do not demonstrate in-situ optimization at this scale, despite claiming in the formalism section (page 3) that their protocol “bypasses” computationally intensive tasks. This ironically raises a question: why not simply optimize in-silico if in-situ optimization is even more challenging? I suggest that the authors clarify the practical limitations and potential trade-offs between in-situ and in-silico implementations.

Our choice to demonstrate iPAC for a smaller scale network was driven by resource constraints rather than a fundamental limitation of our protocol for larger networks. An in-silico approach requires building a highly accurate model of every propagation path, boundary and loss channel in continuous space which can become computationally burdensome and sensitive to modelling errors, especially for complex cavities. Whereas, in-situ, the physical medium computes its own Green’s function in real time, and our protocol only requires local measurements. Furthermore, in our largescale simulations we have mimicked the real-world measurement process by adding noise, suggesting that in-situ optimization is scalable to large network size. We have included these clarifications at the end of the first paragraph of the section “In-silico Implementation of iPAC for large control parameter system”